# Chest X-ray Does Not Predict the Risk of Endotracheal Intubation and Escalation of Treatment in COVID-19 Patients Requiring Noninvasive Respiratory Support

**DOI:** 10.3390/jcm11061636

**Published:** 2022-03-16

**Authors:** Tommaso Pettenuzzo, Chiara Giraudo, Giulia Fichera, Michele Della Paolera, Martina Tocco, Michael Weber, Davide Gorgi, Silvia Carlucci, Federico Lionello, Sara Lococo, Annalisa Boscolo, Alessandro De Cassai, Laura Pasin, Marco Rossato, Andrea Vianello, Roberto Vettor, Nicolò Sella, Paolo Navalesi

**Affiliations:** 1Institute of Anesthesiology and Intensive Care, Padua University Hospital, 13 Via Gallucci, 35121 Padua, Italy; tommasopettenuzzo@gmail.com (T.P.); annalisa.boscolo@gmail.com (A.B.); alessandro.decassai@gmail.com (A.D.C.); laurapasin1704@gmail.com (L.P.); pnavalesi@gmail.com (P.N.); 2Institute of Radiology, Padua University Hospital, 2 Via Nicolò Giustiniani, 35128 Padua, Italy; chiara.giraudo@unipd.it; 3Institute of Anesthesiology and Intensive Care, Department of Medicine, University of Padua, 2 Via Nicolò Giustiniani, 35128 Padua, Italy; michele.dellapao@gmail.com (M.D.P.); mtmartinatocco8@gmail.com (M.T.); marco.rossato@unipd.it (M.R.); roberto.vettor@unipd.it (R.V.); 4Pediatric Radiology, Padua University Hospital, 2 Via Nicolò Giustiniani, 35128 Padua, Italy; gfichera90@gmail.com; 5Department of Biomedical Imaging and Image Guided Therapy, Medical University of Vienna, 23 Spitalgasse, 1090 Vienna, Austria; michael.weber@meduniwien.ac.at; 6Internal Medicine, Department of Medicine, University of Padua, 2 Via Nicolò Giustiniani, 35128 Padua, Italy; davide.gorgi92@gmail.com (D.G.); silviacarlucci@outlook.com (S.C.); 7Respiratory Pathophysiology Division, Department of Cardio-Thoracic, Vascular Sciences and Public Health, University of Padua, 2 Via Nicolò Giustiniani, 35128 Padua, Italy; federico.lionello@aopd.veneto.it (F.L.); sara.lococo@aopd.veneto.it (S.L.); andrea.vianello.1@unipd.it (A.V.)

**Keywords:** chest X-ray, coronavirus disease-19, endotracheal intubation, noninvasive respiratory support

## Abstract

Forms of noninvasive respiratory support (NIRS) have been widely used to avoid endotracheal intubation in patients with coronavirus disease-19 (COVID-19). However, inappropriate prolongation of NIRS may delay endotracheal intubation and worsen patient outcomes. The aim of this retrospective study was to assess whether the CARE score, a chest X-ray score previously validated in COVID-19 patients, may predict the need for endotracheal intubation and escalation of respiratory support in COVID-19 patients requiring NIRS. From December 2020 to May 2021, we included 142 patients receiving NIRS who had a first chest X-ray available at NIRS initiation and a second one after 48–72 h. In 94 (66%) patients, the level of respiratory support was increased, while endotracheal intubation was required in 83 (58%) patients. The CARE score at NIRS initiation was not predictive of the need for endotracheal intubation (odds ratio (OR) 1.01, 95% confidence interval (CI) 0.96–1.06) or escalation of treatment (OR 1.01, 95% CI 0.96–1.07). In conclusion, chest X-ray severity, as assessed by the CARE score, did not allow predicting endotracheal intubation or escalation of respiratory support in COVID-19 patients undergoing NIRS.

## 1. Introduction

Severe acute respiratory syndrome coronavirus 2 (SARS-CoV-2) infection may lead to the development of coronavirus disease-19 (COVID-19). In 5–8% of patients, COVID-19 causes acute hypoxemic respiratory failure (hARF) requiring intensive care unit (ICU) admission [1,2,3,4,5]. Hypoxemic COVID-19 patients often require forms of noninvasive respiratory support (NIRS), i.e., high-flow nasal oxygen (HFNO), continuous positive airway pressure (CPAP), or noninvasive ventilation (NIV), to avoid endotracheal intubation [6,7]. Undue delays in endotracheal intubation may adversely affect patient outcomes and increase mortality because of patient self-inflicted lung injury (P-SILI), i.e., a form of lung injury that depends on the patient’s high respiratory efforts, generating excessively high transpulmonary pressure [8]. The availability of reliable predictors of NIRS failure is therefore of utmost importance to guide the decision to intubate COVID-19 patients receiving NIRS.

Chest X-ray and computed tomography (CT)-based scores quantifying disease severity have been developed and their prognostic accuracy investigated, demonstrating in retrospective studies that they can help predict mortality [9,10,11,12,13,14]. In particular, the CARE score, which is based on the assessment of ground-glass opacity and consolidation at the chest X-ray, was shown to be a predictor of hospital mortality [11]. However, no study has insofar systematically assessed the performance of chest imaging in predicting the risk of endotracheal intubation in COVID-19 hypoxemic patients undergoing NIRS.

In the present study, we aimed to ascertain whether or not the type and extent of chest X-ray abnormalities, as assessed by the CARE score, may help predict the need for endotracheal intubation (primary endpoint) and escalation of respiratory support, the duration of invasive mechanical ventilation, and hospital mortality (secondary endpoints) in patients with hARF secondary to COVID-19.

## 2. Materials and Methods

### 2.1. Patients and Measurements

We included all patients referred to the University Hospital of Padua (Italy), from December 2020 to May 2021, for hARF secondary to COVID-19, who underwent NIRS and had a first chest X-ray available immediately prior to NIRS initiation (first chest X-ray) and a second one after 48–72 h (second chest X-ray) of NIRS. Patients were enrolled from one ICU and two high-dependency units. We excluded patients receiving conventional oxygen therapy (e.g., nasal prongs, simple face masks, venturi mask, non-rebreather mask) as the maximum level of respiratory support, those intubated without receiving NIRS, and those for whom NIRS was the ceiling of treatment. The escalation of respiratory support was defined as any increase in the level of support, i.e., from HFNO to CPAP/NIV or from CPAP to NIV. Among the ICU patients, patients receiving NIRS out of the ICU were excluded. The indications for NIRS initiation, escalation of respiratory support, and endotracheal intubation followed regional guidelines [15]. The study was conducted in accordance with the Declaration of Helsinki and approved by the institutional review board (protocol 183n/AO/21). Patients who survived gave their informed consent for inclusion, whereas those who died were included with a waiver of consent.

Patients were analyzed in two separate groups, according to whether or not they required endotracheal intubation because of NIRS failure. The clinical and laboratory variables collected at NIRS initiation and the outcome variables are described in Appendix A. Arterial blood gas analysis was always obtained right before NIRS initiation.

The CARE score was independently calculated by two radiologists who are experts in thoracic imaging, blinded to patients’ outcomes, and the final score was agreed upon after consensus. The composition and computation of the CARE score have been previously described [11]. Briefly, each lung was divided into three areas (upper, middle, and lower), and a four-grade score separately assessing the extent of ground-glass opacity and consolidation was calculated. In each area, the ground-glass sub-score was graded from 0 to 3 and the consolidation sub-score from 4 to 6. The global CARE score was derived from the sum of the two sub-scores up to a maximum value of 36 [11]. The CARE score was calculated for the chest X-ray prior to NIRS initiation (first CARE score) and the chest X-ray at 48–72 h (second CARE score), and the delta CARE score was computed as the difference between the first and second CARE scores.

The reporting of the present study followed the “Strengthening the reporting of observational studies in epidemiology” (STROBE) statement guidelines (Appendix A) [16].

### 2.2. Statistical Analysis

Baseline variables (i.e., demographic characteristics, comorbidities, and laboratory findings at enrollment) and outcome variables are shown in Appendix A. Categorical variables are presented as absolute numbers (n) and percentages (%). For continuous variables, the median and interquartile range (IQR) are reported. Fisher’s exact test was applied for categorical variables, whereas Wilcoxon’s rank-sum test was used for continuous variables. Wilcoxon’s signed-rank test was used to investigate if the CARE score changed between the first and the second chest X-ray.

The association between the first CARE score and the need for endotracheal intubation was investigated with univariable logistic regression. Multivariable logistic regression was used to assess potential confounders among the baseline variables described in Appendix A. Variables found to be significantly associated with the outcome (*p* < 0.05) were entered into the multivariable model. Multicollinearity was defined as a variance inflation factor (VIF) = 1/(1 − R^2^) greater than or equal to 2.5, where R is the percentage of variance in the individual covariates, and variables characterized by multicollinearity were sequentially removed starting from the variable associated with the highest VIF [17].

As secondary outcomes, multivariable logistic and linear regressions were applied to assess the association between the first CARE score and the escalation of respiratory support, invasive mechanical ventilation duration, and hospital mortality, as appropriate.

All statistical tests were two-tailed, and statistical significance was defined as *p* < 0.05. Statistical analysis was performed using SPSS version 27.0 (SPSS Software, IBM Corp., Armonk, NY, USA) and R version 4.1.1 (R Project for Statistical Computing, Vienna, Austria).

## 3. Results

Of 386 patients admitted in the study period, 142 patients met the inclusion criteria and were analyzed (Figure 1). Patients’ baseline characteristics are reported in Table 1. Patients were 69 (58–75) years old on average, and 44 (31%) patients were female. Patients were admitted after 6 (4–9) days from symptom onset. The Charlson comorbidity index was 3 (2–5). The sequential organ failure assessment (SOFA) score was 3 (2–4), whereas the respiratory component of SOFA was 2 (2–2).

In 83 (58%) patients, NIRS failed and endotracheal intubation was required. As reported in Table 1, we observed a significantly longer duration of symptoms (7 (4–10) vs. 6 (3–8) days, *p* = 0.04), higher SOFA scores (3 (2–4) vs. 2 (2–3), *p* < 0.01), C-reactive protein (CRP) (113 (62–180) vs. 90 (41–123) mg/L, *p* = 0.04), leukocyte counts (7.81 (5.98–11.26) vs. 6.84 (3.32–9.60) × 10^9^ cells/L, *p* = 0.03), and interleukin-6 (67 (39–165) vs. 51 (26–99) pg/mL, *p* = 0.03), and a lower arterial partial pressure of oxygen-to-inspired oxygen fraction ratio (PaO_2_/FiO_2_) (104 (78–134) vs. 148 (105–177) mmHg, *p* < 0.01) at NIRS initiation in the group of patients who were intubated.

Patients’ outcomes are described in Table 2. In 35 (25%) patients, the level of respiratory support was increased. In the group of patients necessitating intubation, prone positioning was overall more frequent (71 [86%] vs. 14 [24%] patients, *p* < 0.01), hospital length of stay was longer (29 (21–41) vs. 16 (12–22) days, *p* < 0.01), and hospital mortality was greater (19 [23%] vs. 1 [2%], *p* < 0.01).

Chest X-rays from two representative patients with similar first CARE scores are presented in Figure 2. The first CARE score was 9 (6–14) and ranged from 0 to 35, while the second decreased to 8 (4–14, *p* = 0.04) and ranged from 0 to 27 (Table 3). The reduction in the CARE score between the first and the second chest X-ray achieved statistical significance for patients receiving intubation, but not for those receiving NIRS (Table 3).

In the univariable logistic regression, the first CARE score (odds ratio (OR) 1.01, 95% confidence interval (CI) 0.96–1.06) was predictive of endotracheal intubation. In the multivariable logistic regression, a lower Charlson comorbidity index (OR 0.79, 95% CI 0.65–0.95, *p* = 0.01) and PaO_2_/FiO_2_ (OR 0.99, 95% CI 0.98–1.00, *p* = 0.01) and a greater CRP (OR 1.01, 95% CI 1.00–1.01, *p* = 0.03) were the only independent predictors of endotracheal intubation (Table 4).

A lower PaO_2_/FiO_2_ (OR 0.99, 95% CI 0.99–1.00, *p* < 0.01) was the only independent predictor of the escalation of respiratory support (Appendix A). The SOFA score was the only predictor of the duration of invasive mechanical ventilation (estimate 1.66, 95% CI 0.16–3.16, *p* = 0.03) (Appendix A). Age and the occurrence of endotracheal intubation were the only independent predictors of hospital mortality in the multivariable logistic regression analysis (OR 1.14, 95% CI 1.03–1.26, *p* = 0.01, and OR 33.50, 95% CI 3.49–122.00, *p* < 0.01) (Appendix A).

## 4. Discussion

We found that the CARE score did not predict the need for endotracheal intubation and escalation of respiratory support in patients receiving NIRS for aHRF secondary to COVID-19. In addition, the CARE score was not predictive of the duration of invasive mechanical ventilation or of hospital mortality.

In COVID-19 patients, NIRS has been proposed to avoid intubation and provide early post-extubation respiratory support [6,7,18]. However, the potential risk of P-SILI may outweigh the benefits of NIRS (e.g., reduced risk of ventilator-associated pneumonia, sedation-related adverse effects) [19,20].

Several clinical risk stratification tools have been developed to predict COVID-19 progression, thereby identifying the required intensity of treatment and the proper setting of care [21,22,23,24]. Radiological techniques assessing COVID-19-related lung abnormalities and based on chest X-rays [9,10,11,12], CT scans [13,14], and lung ultrasounds [25,26,27,28] have been proposed to predict mortality. In particular, the CARE score demonstrated good accuracy in predicting hospital mortality in a previous study including 175 patients overall (44 at home from the emergency department, 95 treated only in the ward, and 36 transferred from the ward to the ICU) [11].

Our results are in keeping with the study of Sargent et al., who found CXR to not be predictive of a composite outcome including intubation or mortality in 58 patients receiving CPAP [10], and consistent with the findings of Bellani et al., who reported greater CRP and a lower PaO_2_/FiO_2_ to be associated with an increased risk of intubation or death in 909 patients receiving CPAP or NIV outside the ICU [29]. Different from our results, in a retrospective cohort study including 140 patients, Xiao et al. observed that the severity of chest X-rays at admission independently predicted the time to intubation in COVID-19 patients admitted to the medical ward [30]. However, only 13% of those patients were receiving supplemental oxygen at the time of the chest X-ray, which suggests those patients definitely had less severe lung involvement.

Differences in patient populations may explain these findings. Patients who require NIRS are, in principle, more severely affected by COVID-19 pneumonia, as characterized by extensive ground-glass opacities and/or consolidations, leading, in general, to more uniformly severe radiological scores. Therefore, the choice to proceed to endotracheal intubation is more likely to depend on factors other than radiological characteristics, such as patients’ clinical status and gas exchange derangements. Additionally worth mentioning is that the median 24 h CARE score in the present study was 9, much higher than the median value of 3 observed in the previous study of Giraudo and colleagues, performed on a population of patients with variable severity [11]. Future studies may apply artificial intelligence techniques to better discriminate different radiological severities on CXR [31].

We found NIRS failure to be an independent risk factor for mortality. This is in keeping with the results of Grasselli et al., who observed that ICU patients who failed NIV had a significantly lower chance of survival as compared to those who did not [32].

Our study has some limitations. First, we cannot exclude selection biases consequent to the retrospective data collection and doubts about generalizability because of the single-center design. Second, the results obtained with a score based on chest X-rays may not apply to CT scans and lung ultrasounds, which might be necessary to define the severity of lung lesions in this patient population [13,14,25,27]. Nonetheless, plain film X-ray is the standard technique in critically ill COVID-19 patients [33]. Third, the interobserver agreement was not evaluated. Nonetheless, the CARE score has already been validated [11,34,35,36], and the chest X-ray analysis was performed by two independent radiologists who were blinded to each other and to patients’ clinical status. Last, we cannot exclude that having included patients infected with different viral variants impacted our findings. Indeed, a recent CT-based study demonstrated a more consolidative pattern in infections due to the strain isolated in South Africa than in those consequent to the European strain [37]. Nonetheless, we are led to believe that the simultaneous occurrence of different variants was unlikely and, in any case, limited in our population, since the present study took place in a relatively short interval of time (6 months).

In conclusion, we observed that a validated chest X-ray severity score did not predict the need for endotracheal intubation and escalation of respiratory support in COVID-19 patients undergoing noninvasive respiratory support.

## Figures and Tables

**Figure 1 jcm-11-01636-f001:**
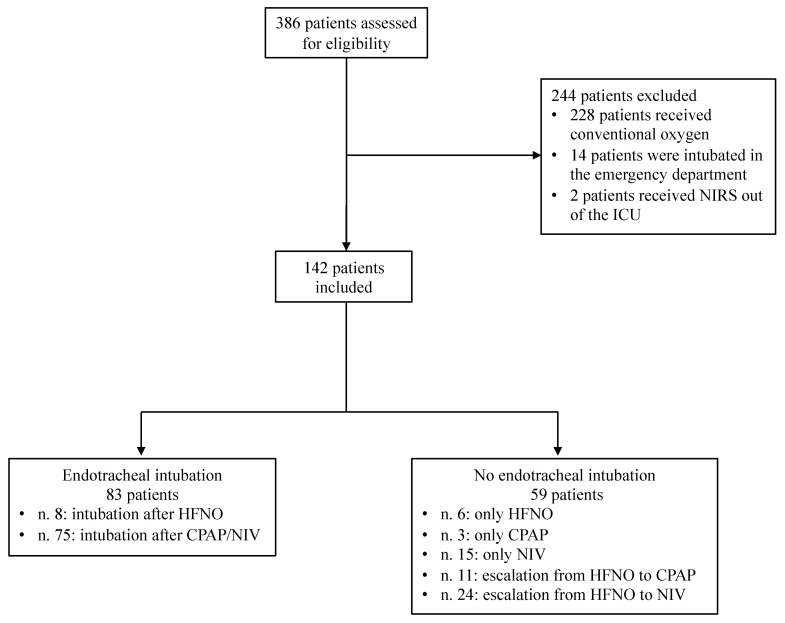
Patient selection flowchart. Abbreviations: NIRS, noninvasive respiratory support; ICU, intensive care unit; HFNO, high-flow nasal oxygen; CPAP, continuous positive airway pressure; NIV, noninvasive ventilation.

**Figure 2 jcm-11-01636-f002:**
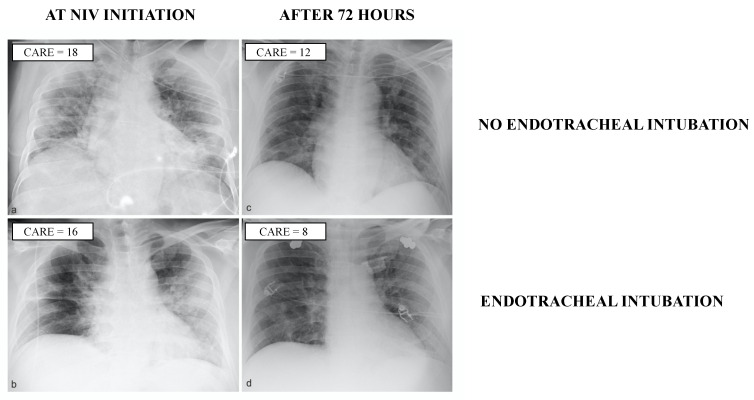
Chest X-rays at noninvasive ventilation (NIV) initiation (**a**,**b**) and after 72 h (**c**,**d**) in two representative male patients (53 years old in (**a**,**c**) and 57 years old in (**b**,**d**)) affected by acute hypoxemic respiratory failure secondary to coronavirus disease-19. Although the two patients had similar CARE scores at NIV initiation and both showed an improvement in the score 72 h after the onset of noninvasive ventilation, only one patient received endotracheal intubation (after 2 days).

**Table 1 jcm-11-01636-t001:** Patients’ baseline characteristics.

Variable	All Patients (n = 142)	No Intubation (n = 59)	Intubation (n = 83)	*p*-Value
Age (years)	69 (58–75)	70 (60–79)	69 (58–73)	0.09
Weight (kg)	78 (69–97)	76 (68–96)	79 (72–102)	0.43
Body mass index (kg/m^2^)	26 (22–31)	25 (22–32)	27 (24–30)	0.66
Female gender (n [%])	44 (31)	19 (32)	25 (30)	0.86
Hypertension (n [%])	81 (57)	35 (59)	46 (55)	0.86
Obesity (n [%])	45 (32)	14 (24)	31 (37)	0.10
Diabetes (n [%])	38 (27)	19 (32)	19 (23)	0.26
Days since symptom onset	6 (4–9)	6 (3–8)	7 (4–10)	0.04
SOFA score	3 (2–4)	2 (2–3)	3 (2–4)	<0.01
Charlson comorbidity index	3 (2–5)	3 (2–5)	3 (2–4)	0.10
C-reactive protein (mg/L)	97 (58–160)	90 (41–123)	113 (62–180)	0.04
Procalcitonin (μg/L)	0.18 (0.06–0.48)	0.13 (0.06–0.48)	0.19 (0.07–0.47)	0.56
D-dimer (μg/L)	323 (171–670)	294 (150–523)	335 (200–801)	0.20
Leukocyte count (× 10^9^ cells/L)	7.58 (4.84–10.57)	6.84 (3.32–9.60)	7.81 (5.98–11.26)	0.03
Lymphocyte count (× 10^9^ cells/L)	0.80 (0.55–1.11)	0.78 (0.48–1.22)	0.80 (0.59–1.10)	0.75
IL-6 (pg/mL)	55 (31–148)	51 (26–99)	67 (39–165)	0.03
PaO_2_/FiO_2_ (mmHg)	118 (90–160)	148 (105–177)	104 (78–134)	<0.01
PaCO_2_ (mmHg)	35 (31–38)	35 (30–38)	35 (31–38)	0.75

Data are reported as the median (interquartile range) or number (percentage), as appropriate. Wilcoxon’s rank-sum test and Fisher’s exact test were applied, as appropriate. Abbreviations: SOFA, sequential organ failure assessment; IL-6, interleukin-6; PaO_2_/FiO_2_, arterial partial pressure of oxygen-to-inspired oxygen fraction ratio; PaCO_2_, arterial partial pressure of carbon dioxide.

**Table 2 jcm-11-01636-t002:** Patients’ outcomes.

Variable	All Patients (n = 142)	No Intubation (n = 59)	Intubation (n = 83)	*p*-Value
Pronation (n [%])	85 (60)	14 (24)	71 (86)	<0.01
Duration of invasive mechanical ventilation (days)	n.a.	n.a.	8 (6–13)	n.a.
Hospital length of stay (days)	22 (14–32)	16 (12–22)	29 (21–41)	<0.01
Hospital mortality (n [%])	20 (14)	1 (2)	19 (23)	<0.01

Data are reported as the median (interquartile range) or number (percentage), as appropriate. Wilcoxon’s rank-sum test and Fisher’s exact test were applied, as appropriate. Abbreviations: n.a., not appropriate.

**Table 3 jcm-11-01636-t003:** The CARE score.

CARE Score	All Patients (n = 142)	No Intubation (n = 59)	Intubation (n = 83)	*p*-Value
First CARE score	9 (6–14)	10 (6–13)	9 (5–15)	0.98
Second CARE score	8 (4–14) *	10 (5–17)	8 (3–12) *	0.04
Delta CARE score	−1 (−5–3)	−1 (−4–6)	−2 (−6–2)	0.01

Data are reported as the median (interquartile range) or number (percentage), as appropriate. Wilcoxon’s rank-sum test, Fisher’s exact test, and Wilcoxon’s signed-rank test were applied, as appropriate. The delta CARE score is the difference between the first and the second CARE score. * *p* < 0.05 from Wilcoxon’s signed-rank test assessing the change in the CARE score between the first and the second chest X-ray.

**Table 4 jcm-11-01636-t004:** Logistic regression for endotracheal intubation.

Variable	Univariable		Multivariable	
	OR (95% CI)	*p*-Value	OR (95% CI)	*p*-Value
First CARE score	1.01 (0.96–1.06)	0.69		
Age	0.97 (0.94–1.00)	0.07		
Female gender	0.91 (0.44–1.86)	0.79		
Days since symptom onset	1.09 (1.00–1.20)	0.06		
SOFA score	1.55 (1.15–2.10)	<0.01	1.40 (0.99–1.99)	0.06
Charlson comorbidity index	0.86 (0.75–1.00)	0.04	0.79 (0.65–0.95)	0.01
C-reactive protein	1.01 (1.00–1.01)	0.04	1.01 (1.00–1.01)	0.03
Procalcitonin	1.06 (0.92–1.22)	0.40		
D-dimer	1.00 (1.00–1.00)	0.66		
Leukocyte count	1.06 (0.99–1.14)	0.88		
Lymphocyte count	0.81 (0.60–1.09)	0.17		
IL-6	1.00 (1.00–1.01)	0.11		
PaO_2_/FiO_2_	0.99 (0.98–1.00)	<0.01	0.99 (0.98–1.00)	0.01
PaCO_2_	1.03 (0.98–1.09)	0.22		

Abbreviations: OR, odds ratio; CI, confidence interval; SOFA, sequential organ failure assessment; IL6, interleukin-6; PaO_2_/FiO_2_, arterial partial pressure of oxygen-to-inspired oxygen fraction ratio; PaCO_2_, arterial partial pressure of carbon dioxide. The variance inflation factors were 1.12 for the SOFA score, 1.13 for the Charlson comorbidity index, 1.02 for C-reactive protein, and 1.11 for PaO_2_/FiO_2_.

## Data Availability

The datasets used and analyzed during the current study are available from the corresponding author on reasonable request.

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
