# Peer review of "Chest X-ray Does Not Predict the Risk of Endotracheal Intubation and Escalation of Treatment in COVID-19 Patients Requiring Noninvasive Respiratory Support"

_jcm, 2022, doi:10.3390/jcm11061636_

Round 1

Reviewer 1 Report

thank you for inviting me to review this interesting paper dedicated to assessment of the risk of endotracheal intubation and escalation of treatment in COVID-19 patients requiring noninvasive respiratory support.

I have only one proposal:

to extend the discussion! it is very short.

Author Response

Reviewer n. 1

Thank you for inviting me to review this interesting paper dedicated to assessment of the risk of endotracheal intubation and escalation of treatment in COVID-19 patients requiring noninvasive respiratory support.

C1: I have only one proposal:

to extend the discussion! It is very short.

R1: Thank you for appreciating our work. As suggested, we extended the discussion to provide the reader with some hypotheses as to why the radiological score could not predict the risk of endotracheal intubation and escalation of treatment and give further insights into future research perspectives. The discussion now reads: “Differences in patient population may explain these findings. Patients who require NIRS are in principle more severely affected by COVID-19 pneumonia, as characterized by extensive ground-glass and/or consolidations, leading, in general, to more uniformly severe radiological scores. Therefore, the choice to proceed to endotracheal intubation is more likely to depend on factors other than radiological characteristics, such as patient clinical status and gas exchange derangements. Also, worth mentioning, the median 24-hour CARE score in the present study was 9, much higher than the median value of 3 observed in the previous study of Giraudo and colleagues, performed on a population of patients with variable severity.11 Future studies may apply artificial intelligence techniques to better discriminate different radiological severity on CXR.31

We found NIRS failure to be an independent risk factor for mortality. This is in keeping with the results of Grasselli et al, who observed that ICU patients who failed NIV had significantly lower chance of survival as compared to those who did not.32

Reviewer 2 Report

Abstract: 

-Line 21: Suggest using the phrasing "The aim of this retrospective study is to assess whether...."

Introduction: 

-Line 43: Consider providing a short description of what P-SILI is. This was a term that isn't commonly used and required me to look it up. 

Methods: 

-Line 64: I am not familiar with the term "high dependency units". Is this a high level of care similar to an ICU or stepdown unit? 

-Line 64: Were any patients who were on supplemental oxygen excluded or only those who were on supplemental oxygen at the time of the first x-ray? I would imagine most patients requiring NIRS also required supplemental oxygen prior to this. Were these patient excluded or not. Please clarify

-Line 66: You excluded "those for whom NIRS was ceiling of treatment", but 59 patient's included in the study were not intubated (and therefore NIRS was the ceiling of treatment. Please clarify.

 -Line 71: Although this was a retrospective study, each patient was consented for inclusion in the study after the fact? How was this handled for those who had in hospital mortality?

-Line 78: The CARE score was previously validated and is discussed here, but no mention of the component scores (CO/GG) are described in the methods. One part to mention is the CO subscore in the original paper was found to be significant but the GG was not. This  begs the question of why would the subscores be evaluated if only the CARE score and CO subscore were deemed significant in the validation study. Consider simplifying your results and reporting on the the overall CARE scores. 

Results: 

-Table 3: See above statements regarding CO and GG subscores 

Discussion: 

-Line 191: One additional limitation could be due to the variety of COVID-19 variants which may have impacted the present study compared to the variant in the initial CARE score validation. 

-Although this is a negative study, I think there can be more said about what future avenues for research related to this project could exist. I think you provide a nice summary of the problem and description of limitations. 

Author Response

Reviewer #2

C1: Abstract:

-Line 21: Suggest using the phrasing "The aim of this retrospective study is to assess whether...."

R1: Taken.

C2: Introduction:

-Line 43: Consider providing a short description of what P-SILI is. This was a term that isn't commonly used and required me to look it up.

R2: We added a definition for P-SILI in the Introduction, which now reads: “Undue delays in endotracheal intubation may adversely affect patient outcome and increase mortality because of patient self-inflicted lung injury (P-SILI), i.e., a form of lung injury that depends on the patient's high respiratory efforts, generating excessively high transpulmonary pressure.”

C3: Methods:

-Line 64: I am not familiar with the term "high dependency units". Is this a high level of care similar to an ICU or stepdown unit?

R3: As you mentioned, “high dependency unit” is a synonym for “stepdown unit”, i.e., wards for patients requiring more intensive observation, treatment, and care than a general ward but less than an intensive care unit. We are available to use the phrasing “stepdown unit” if you believe this is more appropriate.

C4: -Line 64: Were any patients who were on supplemental oxygen excluded or only those who were on supplemental oxygen at the time of the first x-ray? I would imagine most patients requiring NIRS also required supplemental oxygen prior to this. Were these patients excluded or not? Please clarify

R4: Patients were excluded if they were on conventional oxygen as the maximum level of respiratory support and never received NIRS. We amended the manuscript accordingly: “We excluded patients receiving conventional oxygen therapy (e.g., nasal prongs, simple face masks, Venturi mask, non-rebreather mask) as the maximum level of respiratory support, those intubated [...]”

C5: -Line 66: You excluded "those for whom NIRS was ceiling of treatment", but 59 patient's included in the study were not intubated (and therefore NIRS was the ceiling of treatment. Please clarify.

R5: We defined ceiling of treatment as any limitation in vital organ support that was predetermined because of the patient’s wishes, values, beliefs, or clinical conditions. Those 59 patients who were not intubated did not require intubation because their clinical conditions improved during the disease course. We are available to modify the manuscript to better clarify this concept if you deem this is appropriate.

C6:  -Line 71: Although this was a retrospective study, each patient was consented for inclusion in the study after the fact? How was this handled for those who had in hospital mortality?

R6: We thank the Reviewer for allowing us to clarify this. Patients who died did not give their informed consent for inclusion. However, our local Institutional Review Board did not require these patients to be excluded, given the retrospective nature of the study and anonymous data collection. We clarified this in the Methods, which now reads: “The study was conducted in accordance with the Declaration of Helsinki and approved by the Institutional Review Board (protocol 183n/AO/21). Patients who survived gave their informed consent for inclusion, whereas those who died were included with a waiver of consent”.

C7: -Line 78: The CARE score was previously validated and is discussed here, but no mention of the component scores (CO/GG) are described in the methods. One part to mention is the CO subscore in the original paper was found to be significant but the GG was not. This  begs the question of why would the subscores be evaluated if only the CARE score and CO subscore were deemed significant in the validation study. Consider simplifying your results and reporting on the the overall CARE scores.

R7: Thank you for your comment. The components of the CARE were already described in the methods section (page 2 line 82 to 84). The Reviewer is right in highlighting that in the original paper, proposing and validating the CARE, it emerged that only the overall score and the consolidation subscore were predictive of the outcome. Nevertheless, we could have not excluded the GG score from this study for two main reasons: 1) the overall CARE score can be computed only if both components are assessed (i.e., GG and Co); 2) removing the GG subscore, we would have introduced a bias also because the type of assessed population is very different in the two studies (i.e., the current and the study previously published by our group). In fact, in the current project we evaluated only patients undergoing NIRS during the second and third waves of SARS-CoV-2 pandemic, while in the previous study all patients with COVID-19 admitted to our tertiary center during the first month of the outbreak of the pandemic in our area were assessed. Despite this, we understand the point of the reviewer and we’ve embraced his/her suggestion simplifying the results of the CARE score. Thus, we have removed the information of the subscores throughout the manuscript and from table 3 and the supplementary material.

C8: Results:

-Table 3: See above statements regarding CO and GG subscores

R8: Thank you. As previously mentioned, we have removed the subscores from table 3.

C9: Discussion:

-Line 191: One additional limitation could be due to the variety of COVID-19 variants which may have impacted the present study compared to the variant in the initial CARE score validation.

R9: We thank the reviewer for addressing this issue. Certainly, the impact of different variants should be addressed and now we have added one reference [Brakohiapa et al https://doi.org/10.1016/j.heliyon.2021.e07818] and the following sentence in the limitations section “Last, we cannot exclude that having included patients infected by different viral variants has impacted our findings. Indeed, a recent CT-based study demonstrated a more consolidative pattern in infections due to the strain isolated in South Africa than in those consequent to the European strain37. Nonetheless, we are led to believe that the simultaneous occurrence of different variants was unlikely and in any case limited in our population, since the present study took place in a relatively short interval of time (6 months).”

C10: -Although this is a negative study, I think there can be more said about what future avenues for research related to this project could exist. I think you provide a nice summary of the problem and description of limitations.

R10: We thank the Reviewer for appreciating our work. We further developed the discussion as follows: “Differences in patient population may explain these findings. Patients who require NIRS are in principle more severely affected by COVID-19 pneumonia, as characterized by extensive ground-glass and/or consolidations, leading, in general, to more uniformly severe radiological scores. Therefore, the choice to proceed to endotracheal intubation is more likely to depend on factors other than radiological characteristics, such as patient clinical status and gas exchange derangements. Also, worth mentioning, the median 24-hour CARE score in the present study was 9, much higher than the median value of 3 observed in the previous study of Giraudo and colleagues, performed on a population of patients with variable severity.11 Future studies may apply artificial intelligence techniques to better discriminate different radiological severity on CXR.31

We found NIRS failure to be an independent risk factor for mortality. This is in keeping with the results of Grasselli et al, who observed that ICU patients who failed NIV had significantly lower chance of survival as compared to those who did not.32